# Bony-fish-like scales in a Silurian maxillate placoderm

Xindong Cui[1,2], Matt Friedman [3], Yilun Yu[2,4], You-an Zhu [2] ✉ & Min Zhu [2,4] ✉

Major groups of jawed vertebrates exhibit contrasting conditions of dermal plates and scales. But the transition between these conditions remains unclear due to rare information on taxa occupying key phylogenetic positions. The 425-million-year-old fish *Entelognathus* combines an unusual mosaic of characters typically associated with jawed stem gnathostomes or crown gnathostomes. However, only the anterior part of the exoskeleton was previously known for this very crownward member of the gnathostome stem. Here, we report a near-complete post-thoracic exoskeleton of *Entelognathus*. Strikingly, its scales are large and some are rhomboid, bearing distinctive peg-and-socket articulations; this combination was previously only known in osteichthyans and considered a synapomorphy of that group. The presence in *Entelognathus* of an anal fin spine, previously only found in some stem chondrichthyans, further illustrates that many characters previously thought to be restricted to specific lineages within the gnathostome crown likely arose before the common ancestor of living jawed vertebrates.

Modern gnathostomes (jawed vertebrates) include osteichthyans (bony fishes and tetrapods) and chondrichthyans (cartilaginous fishes). These two groups exhibit contrasting conditions in their exoskeletons. Osteichthyans have large (macromeric) dermal plates and scales that are highly variable in shape, whereas chondrichthyans have tiny (micromeric) scales or tesserae that are small and uniform in shape[1]. Most recent phylogenetic analyses imply that the macromeric dermal skeleton is primitive for gnathostomes, given it occurs in jawed stem gnathostomes known as placoderms[2–8]. The transition between the patterns of the macromeric dermal plates of the stem and crown gnathostomes is perhaps best captured by late Silurian taxa called "maxillate placoderms", of which *Entelognathus primordialis* and *Qilinyu rostrata* are the best known examples[9,10]. Showing a typical placoderm-like skull roof and trunk armor in combination with osteichthyan-like marginal jaw bones and cheekbones, these taxa provide strong support for the homology and evolutionary continuity of several components in the macromeric exoskeletons of placoderms and osteichthyans[9,10].

Like the anterior part of the exoskeleton, the scales that make up the posterior part of the exoskeleton also exhibit contrasting morphology and histology across major gnathostome groups. Many early chondrichthyan scales are very small and thick, have a crown and bulging base separated by a constriction or neck, and lack structures for articulation between adjacent scales[11–15]. Crowns in some early chondrichthyan scales are covered by enameloid, and consist of a single odontode (monoodontode) or multiple odontodes (polyodontode)[16–22]. Early osteichthyans, on the other hand, bear large rhomboid scales with a well-developed peg-and-socket articulation[23–28], which is considered a bony fish synapomorphy[29]. Osteichthyan scales generally bear superimposed layers of enamel and orthodentine, and a well-developed vasculature[30–37]. The scales of placoderms vary morphologically, but are typically small, and ornamented with mesodentine or semidentine tubercles[38–40]. The transition in scale conditions between stem-group and crown-group lineages remains unclear, particularly for features of osteichthyans commonly used to identify isolated scales. A lack of evidence from maxillate placoderms, which have been instrumental in resolving parallel

[1]Key Laboratory of Orogenic Belts and Crustal Evolution, School of Earth and Space Sciences, Peking University, 100871 Beijing, China. [2]CAS Key Laboratory of Vertebrate Evolution and Human Origins, Institute of Vertebrate Paleontology and Paleoanthropology, Chinese Academy of Sciences, 100044 Beijing, China. [3]Museum of Paleontology and Department of Earth and Environmental Sciences, University of Michigan, Ann Arbor, MI 48109, USA. [4]University of Chinese Academy of Sciences, 100049 Beijing, China. ✉e-mail: zhuyouan@ivpp.ac.cn; zhumin@ivpp.ac.cn

problems for the cranial dermal skeleton, contributes to this uncertainty[9,10].

Here, we report a near-complete post-thoracic exoskeleton of *Entelognathus* (IVPP V32322) from the Kuanti Formation (late Silurian, ~425 million years ago) of Qujing, Yunnan Province, China. This maxillate placoderm was previously known only from the head and trunk shields[9]. Using micro-computed tomography (μCT), we visualized the squamation and spines preserved in situ, as well as some trunk plates, providing crucial information on the evolution of scales in jawed vertebrates. The post-thoracic exoskeleton of *Entelognathus* reveals a surprising mosaic of scale and fin spine characters previously thought restricted to osteichthyans and chondrichthyans, respectively.

## Results

The articulated specimen (Figs. 1–3 and Supplementary Fig. 1) preserves the nearly complete post-thoracic portion of the body, in association with part of the trunk shield (median dorsal plate and the posterior dorsolateral plate). The preserved scale-covered part is about 9.5 cm in total length. It is flattened along the sagittal plane, with the right part slightly skewed anterodorsally. The fish (Supplementary Fig. 2 and Supplementary Movie 1) has an estimated total length of ~21.0 cm and an estimated maximum depth of ~5.0 cm. The scale-covered part represents just over half (ca. 52%) of the overall body length. The squamation consists of 11 rows of scales on each side (Fig. 4a–c). The lateral line scale row (Ll, Fig. 4a) extends roughly anteroposteriorly in the middle of the body in life position, with 2 to 3 rows (D, Fig. 4a) of scales dorsal to it and 2 to 7 rows (V, Fig. 4a) of scales ventral to it. The number of scales per row decreases with increasing distance from the lateral line scale row. Rows are numbered in sequence based on proximity to the lateral line scale row. Each scale is numbered according to its row and its sequence within the row (Fig. 4a). For example, the first scale in the first row dorsal to the lateral line row is numbered "1D01".

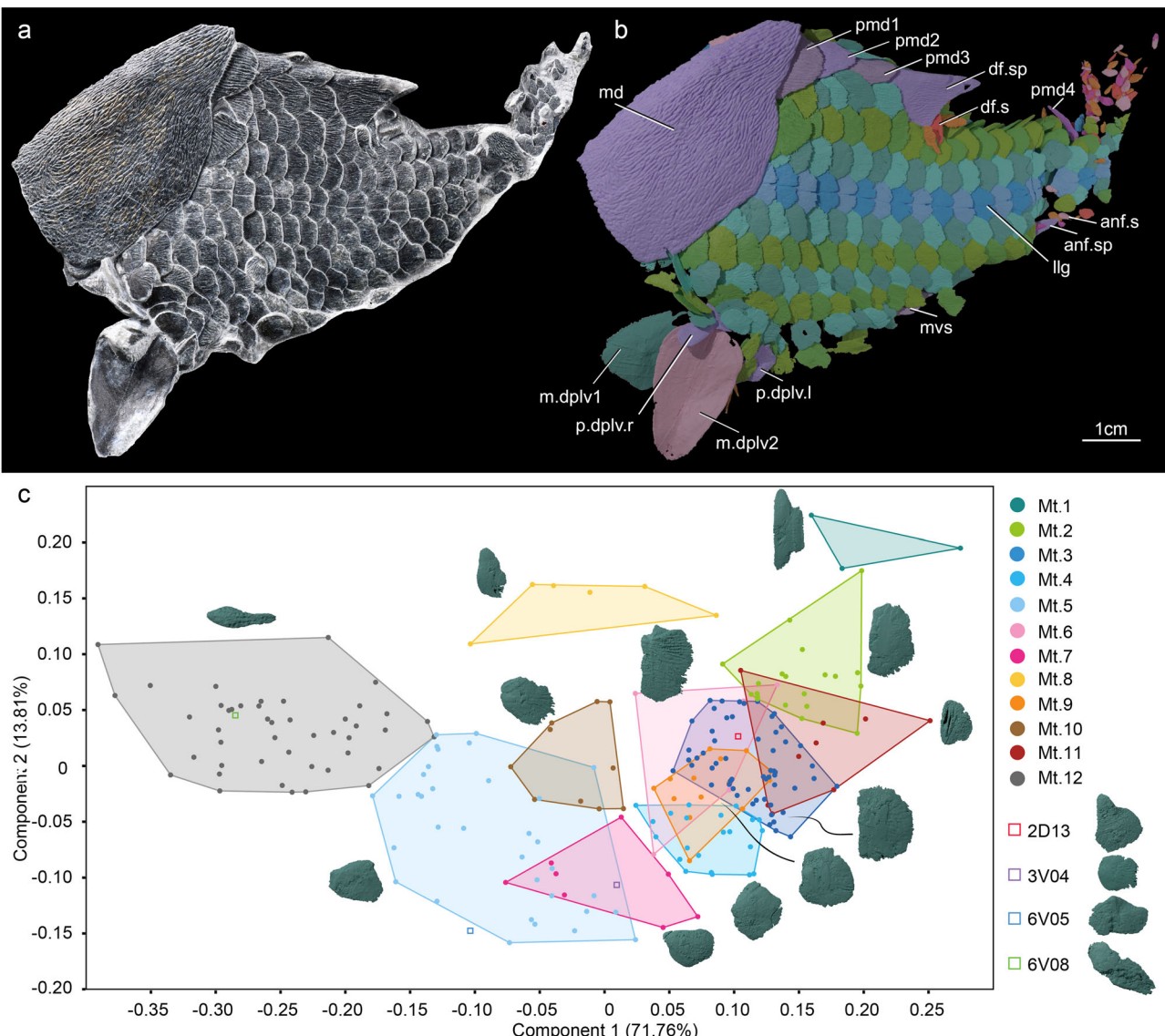

Fig. 1 | *Entelognathus primordialis* (IVPP V32322). **a** Photograph in left lateral view. **b** Virtual rendering in left lateral view. **c** Shape space PCA scatter plot showing PC1 versus PC2 of *Entelognathus* scales (*n* = 237 flank scales) based on landmark-based geometric morphometrics. anf.s anal fin scales, anf.sp anal fin spine, df.s dorsal fin scales, df.sp dorsal fin spine, llg lateral line groove, md median dorsal plate, Mt.1–12, Morphotype 1 to Morphotype 12, mvs median ventral scale, m.dplv1 first median dermal pelvic plate, m.dplv2 second median dermal pelvic plate, pmd1–4 first to fourth postmedian dorsal scales, p.dplv.l left dermal pelvic plate, p.dplv.r right dermal pelvic plate.

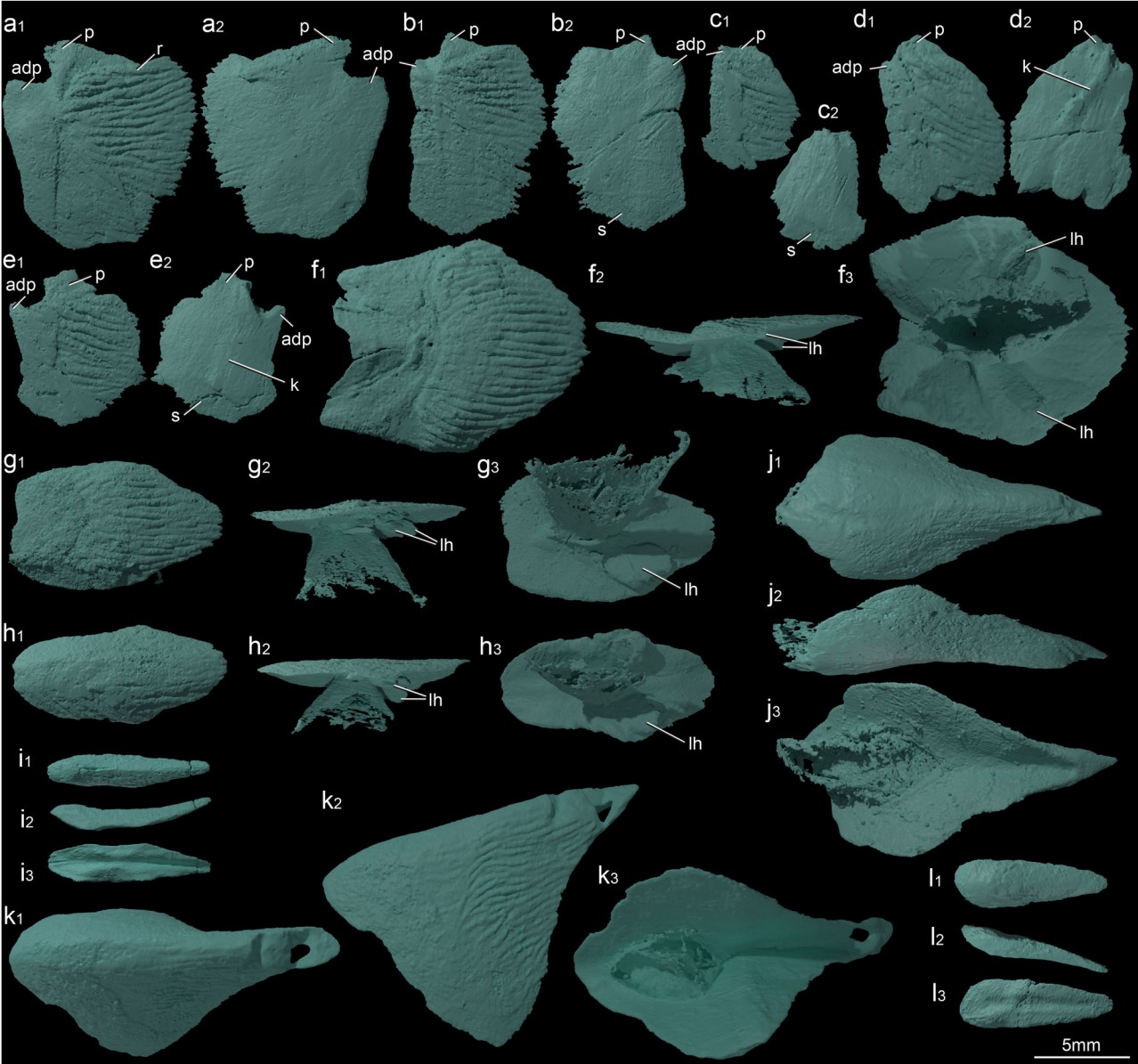

**Fig. 2 | Scales and fin spines of *Entelognathus primordialis* (IVPP V32322).**
**a** Morphotype 6 (2D01) in crown view (**a₁**) and basal view (**a₂**). **b** Morphotype 6 (2D04) in crown view (**b₁**) and basal view (**b₂**). **c** Morphotype 6 (3D02) in crown view (**c₁**) and basal view (**c₂**). **d** Morphotype 6 (3D03) in crown view (**d₁**) and basal view (**d₂**). **e** Morphotype 6 (3D04) in crown view (**e₁**) and basal view (**e₂**). **f** First postmedian dorsal scale in crown view (**f₁**), lateral view (**f₂**), and basal view (**f₃**). **g** Second postmedian dorsal scale in crown view (**g₁**), lateral view (**g₂**), and basal view (**g₃**). **h** Third postmedian dorsal scale in crown view (**h₁**), lateral view (**h₂**), and basal view (**h₃**). **i** Fourth postmedian dorsal scale in crown view (**i₁**), lateral view (**i₂**), and basal view (**i₃**). **j** Median ventral scale in crown view (**j₁**), lateral view (**j₂**), and basal view(**j₃**). **k** Dorsal fin spine in crown view (**k₁**), lateral view (**k₂**), and basal view (**k₃**). **l** Anal fin spine in crown view (**l₁**), lateral view (**l₂**), and basal view(**l₃**). adp anterodorsal process, k keel, lh lingulate humps, p peg, r ridges, s socket.

## Morphology

The flank scales of *Entelognathus* are large and thin, resembling osteichthyan scales rather than the typical small, thick scales of placoderms. They are rounded or polygonal in shape, with parallel longitudinal ridges on the crown (Figs. 2, 3a, and 4a and Supplementary Fig. 3). In crown view, the flank scales bear a prominent concealed field (Figs. 2a–e and 3a and Supplementary Fig. 4a–t) lacking ornamentation, and are covered by more anterior, anterodorsal and anteroventral scales. The posterior, posterodorsal and posteroventral parts of the scale base are slightly depressed (Fig. 2a–e and Supplementary Fig. 4) to accommodate the overlap of adjacent scales. The flank scales can be divided into 12 morphotypes as well as four special scales, based on both discrete morphological differences and results of geometric morphometric analysis (Fig. 1c and Supplementary Fig. 3).

Among the 12 morphotypes, Morphotype 6 (Fig. 2a–e) is particularly noteworthy for its resemblance to the flank scales of osteichthyans. Two features in particular stand out: rhomboid shape and articulations to the median dorsal scales and ventrally adjacent scales via an arrangement evocative of the classic bony fish peg-and-socket arrangement. They also have an anterodorsal part (Fig. 2a–e) of the concealed field that extends dorsally, suggestive of the anterodorsal process of osteichthyans. In addition, the longest axis is slightly oblique to the long axis of the fish. Some scales of Morphotype 6 bear a vertical ridge (Fig. 2d₂, e₂) on the middle of the base, resembling the basal keel of early osteichthyan rhomboid scales. In contrast, scales of other morphotypes do not share a rhomboid shape, peg-and-socket and anterodorsal process articulations, and a basal keel. A detailed description of each

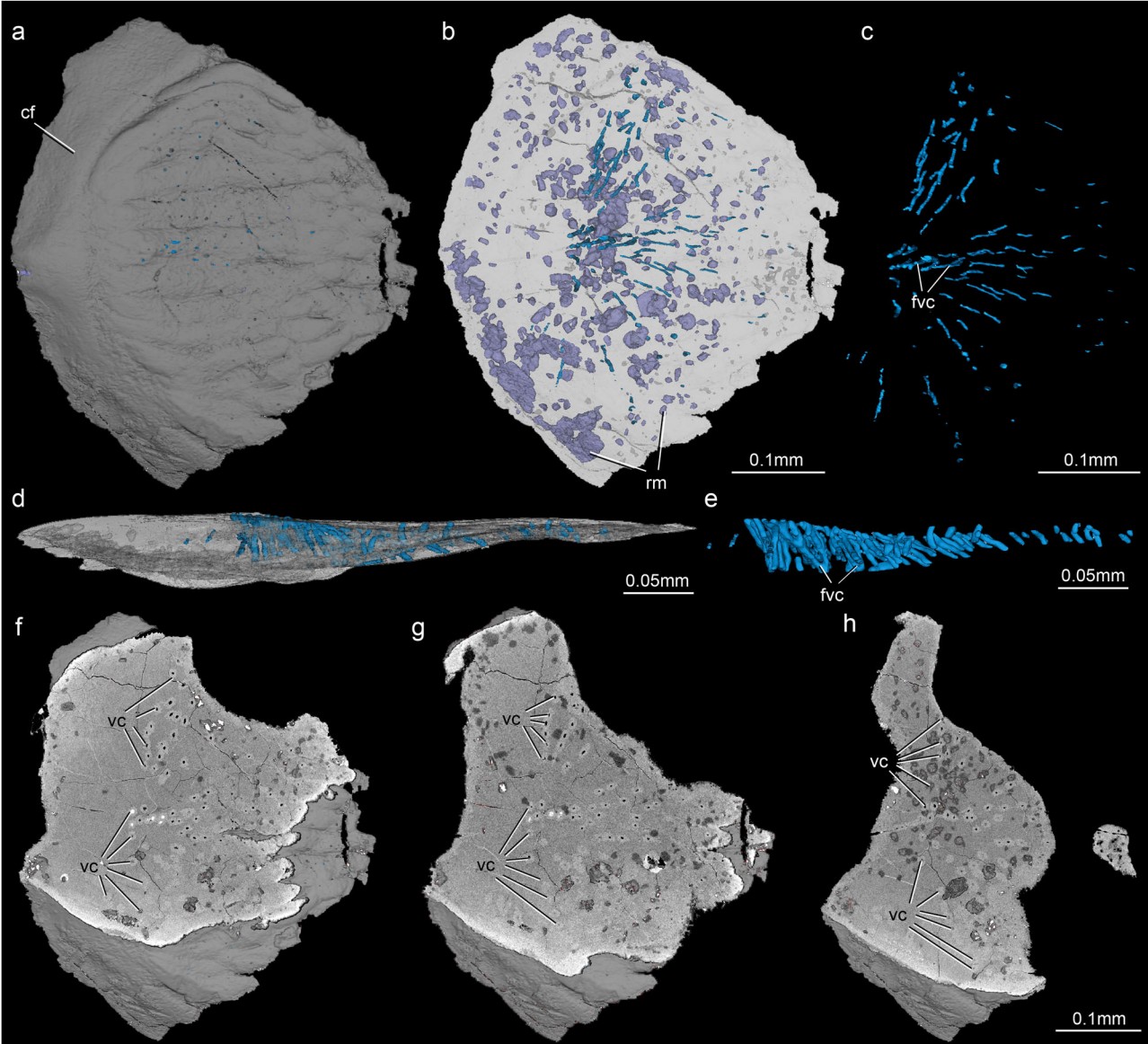

**Fig. 3 | 3D virtual model of a Morphotype 4 scale (V32323.5). a** 3D virtual model in crown view. **b** 3D virtual model with canal system in crown view. **c** Canal system in crown view. **d** 3D virtual model with canal system in lateral view. **e** Canal system in lateral view showing the forked vascular canals. **f–h** CT slices in crown view showing the ascending vascular canals. cf concealed field, rm recrystallized minerals, fvc forked vascular canals, vc vascular canals.

morphotype is provided in the Supplementary Information (Note 1).

The lateral line scales are located in a single row that extends along the middle flank of the fish body (Fig. 1a, b and Supplementary Fig. 1). They are identical to scales that are adjacent ventrally and dorsally, except for a distinct lateral line groove on the crown and associated notch at the posterior edge. The lateral line groove begins at the anterior edge of the ornamented field on the crown and connects posteriorly with the V-shaped notch.

There are three postmedian dorsal plates/scales (pmd1–3, Fig. 1b, median dorsal or ridge scales in osteichthyan terminology) located between the median dorsal plate and the dorsal fin, and one (pmd4, Fig. 1b) near the tail. The first three are roughly equal in length (about 12 mm), but more anterior members of the series are wider than more posterior scales (Fig. 2f–h). The first scale (Fig. 2f) measures 13.35 mm in length and 12.46 mm in width. It has a concave anterior edge, two straight anterior lateral edges, two almost straight posterior lateral edges, and a convex posterior edge. An isosceles-trapezoid-shaped

concealed field is obviously depressed, sitting in front of the heart-shaped free field bearing 26 conspicuous ridges (Fig. 2f₁). A well-developed laterally compressed conical structure radiating out from the center of the base is apparent in lateral and basal views (Fig. 2f₂, f₃). This is identical to the keel-like structure of the median dorsal plate. In addition, the base of the scale (Fig. 2f₃) bears two lingulate humps that extend postero-ventrolaterally, and articulate with flank scales (2D04). The second and third postmedian dorsal scales (Fig. 2g, h) are described in the Supplementary Information (Note 2).

The fourth scale is located far posterior to the first three postmedian dorsal scales, at the transition from the body to the tail. It is elongated and rhombic in dorsal view (Fig. 2i₁). In lateral view, its dorsal margin is slightly concave (Fig. 2i₂). The anterior part is swollen, with a slender depressed area along the ventral margin covered by the flank scales. The middle part of the ventral margin is slightly concave (Fig. 2i₂). The free field is ornamented by faint longitudinal straight ridges (Fig. 2i₁). The internal surface of the scale bears a deep midline groove (Fig. 2i₃). The middle part of the internal surface has a crescent-

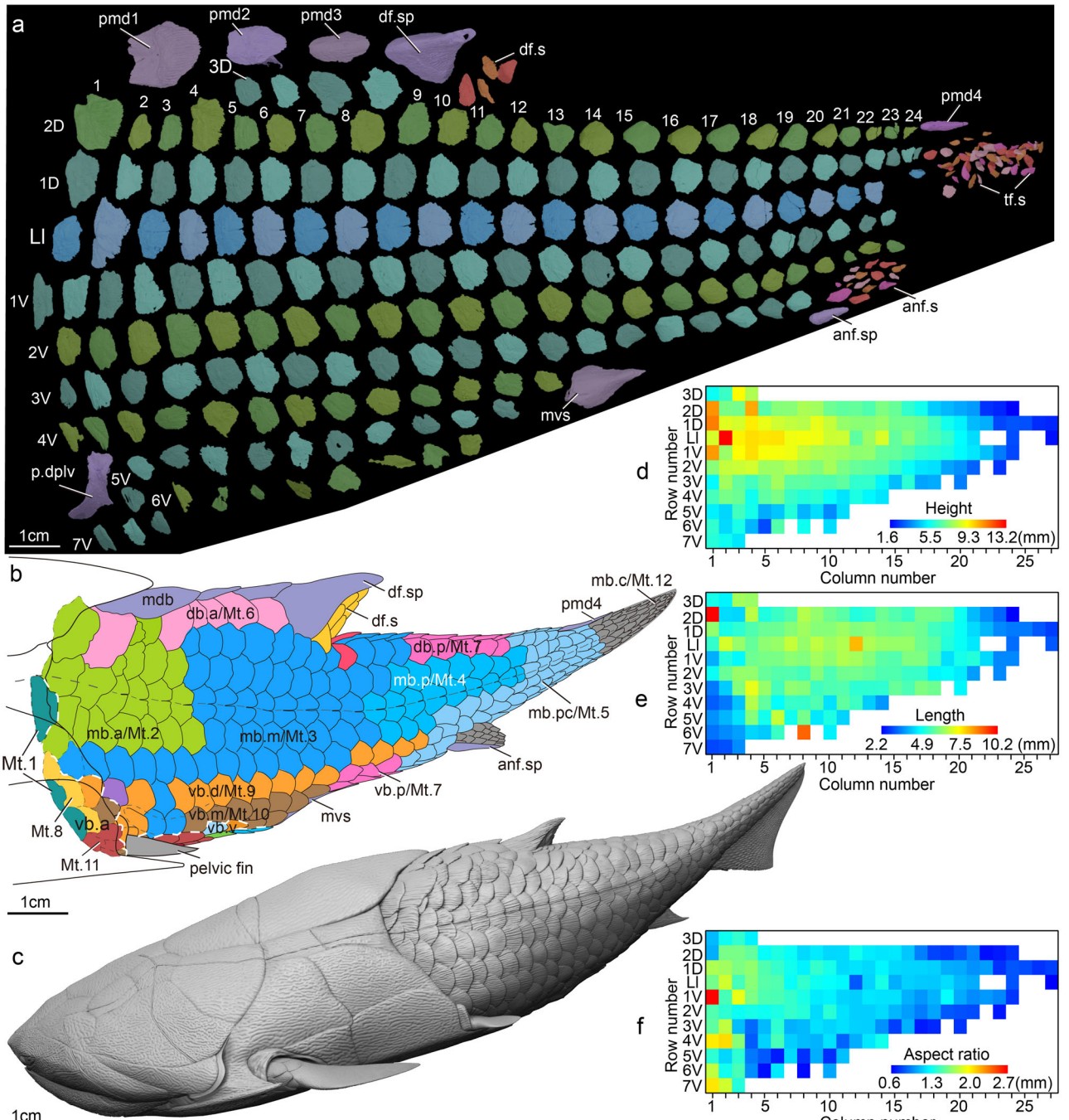

**Fig. 4 | Body plan of *Entelognathus primordialis* (IVPP V32322). a** 3D virtual models of scales and spines in left lateral view. **b** Squamation and body plan in lateral view. **c** Reconstruction of *Entelognathus primordialis* in lateral view. **d** Matrix plot showing the height of flank scales. **e** Matrix plot showing the length of flank scales. **f** Matrix plot showing the aspect ratio of flank scales. anf.s anal fin scales, anf.sp anal fin spine, db.a anterior area of dorsal belt, db.p posterior area of dorsal belt, df.s dorsal fin scales, df.sp dorsal fin spine, Ll lateral line scale row, mb.a anterior area of middle belt, mb.c caudal area of middle belt, mb.m middle area of middle belt, mb.p posterior area of middle belt, mb.pc precaudal area of middle belt, mdb median dorsal belt, Mt.1–12, Morphotype 1 to Morphotype 12, mvs median ventral scale, pmd1–4 first to fourth postmedian dorsal scales, p.dplv dermal pelvic plate, tf.s tail fin scales, vb.a anterior area of ventral belt, vb.d dorsal area of ventral belt, vb.m middle area of ventral belt, vb.p posterior area of ventral belt, vb.v ventral area of ventral belt, 1–24 column number of the scales, 1D–3D first, second and third rows of scales dorsal to the lateral line scale row, 1V–7 V first to seventh rows of scales ventral to the lateral line scale row.

shaped depressed area on each side of the midline groove, covering flank scales.

One median ventral scale (mvs, Fig. 4a) is located immediately behind the flank scale 4V13. This kite-shaped scale has a bulged anterior part and a tapering posterior part defining a sharp posterior tip (Fig. 2j). In external view, the anterior and anterolateral margins bear a smooth concealed field covered by the adjacent flank scales.

Longitudinal ridges ornament the free field. The internal surface shows an oval keel or insertion (Fig. 2j$_3$) anteriorly, similar to that of the postmedian dorsal scales. Unlike the latter, the median ventral scale lacks the lingulate humps on the internal side.

One dorsal fin spine and one anal fin spine are preserved in situ with the squamation (Figs. 1a, b, 2k, l, and 4a–c). These spines are short and robust, with a distinctive tip protruding away from the body

surface, and followed by small fin scales. These features allow for confident identification as spines like those in early osteichthyans (dorsal fin spine) and stem chondrichthyans (dorsal and anal fin spines).

The dorsal fin spine is kite-shaped in dorsal view (Fig. $2k_1$) and triangular in its lateral profile (Fig. $2k_2$), with strong lateral compression in the posterior portion, resulting in a V-shaped cross-section (Fig. $2k_3$). The anterior edge of the external surface is smooth but raised, covered by the third postmedian dorsal scale anteriorly and flank scales laterally. By contrast, the free field bears longitudinal wavy ridges. The internal surface (Fig. $2k_3$) is smooth except for an oval insertion area similar to that of the postmedian dorsal scales and the median ventral scale. Several narrow fin scales are preserved behind the dorsal fin spine (Fig. 4a–c).

The anal fin is present, indicated by the anal fin spine and associated small fin scales (Fig. 4a–c). The anal fin spine is small and slender (Fig. 2l). Its external surface (Fig. $2l_1$) is convex, whereas its internal surface (Fig. $2l_3$) is concave with a deep midline groove like that of the fourth postmedian dorsal scale (Fig. $2i_3$).

### Histology

To investigate the histology of *Entelognathus*, we selected an anterior ventrolateral plate, an incomplete posterior lateral plate, and two scales to make thin sections (Supplementary Fig. 5a–d). We also sampled a lower jaw and one scale of the Silurian osteichthyan *Guiyu oneiros* (Supplementary Fig. 5e, f) for comparison. In addition, we scanned one scale of *Entelognathus* using μCT to reconstruct the internal canal system in three dimensions (Fig. 3). The dermal bone plates and scales of *Entelognathus* display similarities in histology to one another, and they are consistent with those of *Guiyu*. They present two principal divisions: a thin superficial layer and a thick compact basal layer (Supplementary Fig. 5a–d). The fine details and the nature of the superficial layer are obscured due to heavy recrystallization. The basal layer is composed of cellular lamellar bone. The osteocyte lacunae are mainly distributed in the middle and upper parts of the lamellar bone in clusters and inconspicuous lamellar forms (Supplementary Fig. 5a–d). The thin sections show that there are many round and polygonal recrystallized minerals in the superficial and basal layers (Supplementary Fig. 5b–d). The shape and arrangement of the minerals suggest that they are not related to the vasculature.

The 3D virtual model of the scale shows that these minerals are polyhedrons of different sizes (Fig. 3b). These are mainly distributed in the basal layer, some scattered and some gathered together, indicating that they have no relationship with the vasculature (Fig. 3b). However, the 3D virtual model restores the genuine vascular canals of the scale. In the crown view, the canal system consists of scattered ascending canals that are mainly concentrated in the middle and posterior parts of the scale (Fig. 3a–d, f–h). The whole canal system is semicircular, and the ascending canals incline to the center and are arranged in radial lines (Fig. 3b, c, f–h). In the lateral view, the ascending canals appear at the same level, but they become shorter from anterior to posterior (Fig. 3d, e). These canals extend through the superficial and basal layers of the scale. The central canals bifurcate (Fig. 3c, e).

### Squamation

The scale-covered portion of *Entelognathus* is divided into four main belts based on the distribution of the scale morphotypes: the median dorsal, dorsal, middle, and ventral belts. They are further divided into 13 areas corresponding to postmedian dorsal scales and 12 morphotypes of flank scales (Fig. 4b). The median dorsal belt contains three symmetrical unpaired median dorsal scales described in detail above. As in *Parayunnanolepis xitunensis*[41], the unpaired median dorsal scales of *Entelognathus* are mainly confined to the area anterior to the dorsal fin. This differs from *Pterichthyodes milleri*[42], *Kujdanowiaspis podolica*[43], *Xiushanosteus mirabilis*[44], and

some early osteichthyans[28,45,46] where unpaired median dorsal scales also occupy the space behind the dorsal fin. A detailed description of the squamation pattern can be found in the Supplementary Information (Note 3).

As a whole, the scales of *Entelognathus* change in morphology in accordance with their positions (Fig. 4a). The scales in the anterior area of the middle belt (mb.a, Fig. 4b) are the highest, and the surrounding scales become shorter posteriorly, dorsally, and ventrally (Fig. 4d). The lateral line scales in the middle area of the middle belt (mb.m, Fig. 4b) are the longest, and the surrounding scales generally gradually become shorter, with a few scales having a significantly abnormal length (Fig. 4e). The scales in the anterior area of the ventral belt (vb.a, Fig. 4b) and the caudal area of the middle belt (mb.c, Fig. 4b) show distinctly smaller lengths (Fig. 4e). The scales in the anterior area of middle belt (mb.a, Fig. 4b) and the anterior area of the ventral belt (vb.a, Fig. 4b) have a larger aspect ratio that decreases posteriorly (Fig. 4f).

### Phylogenetic analysis

To investigate the impact of the updated character combination of *Entelognathus* on its systematic placement and character evolution in early gnathostomes, we conducted a phylogenetic analysis using a dataset with 694 characters and 159 taxa (Supplementary Data 1 and Supplementary Notes 4 and 5). The dataset was mainly based on the work of Zhu et al.[44] with the addition of four scale characters (Character 691, shape of the trunk scales; Character 692, dermal ornament with parallel vermiform ridges on the trunk scales; Character 693, dermal ornament with concentric ridges on the trunk scales; Character 694, dermal ornament with tubercles on the trunk scales) and one duplicated character (original Character 450, repeated with Character 448) was deleted. Some character codings were checked, updated or revised, which are marked in light blue in Supplementary Data 1. The phylogenetic analysis placed *Entelognathus* and *Qilinyu* in a clade as the immediate sister lineage of crown gnathostomes, confirming both the pivotal position and the monophyly of maxillate placoderms (Fig. 5 and Supplementary Fig. 6). The support for the latter varies among previous analyses[9,10,22,44,47–49], possibly due to the significant morphological variation between the two taxa. The post-thoracic morphology, scales, and histology of the exoskeleton in other maxillate placoderms remain unclear due to a lack of relevant fossil material[10,49].

## Discussion

The scales and squamation of *Entelognathus* show striking similarities to those of osteichthyans, in addition to the osteichthyan-like marginal jaw and cheek plates[9]. This provides strong evidence that distinctive characters of osteichthyan scales, such as the peg-and-socket articulation, actually arose on the gnathostome stem lineage. In contrast to the thick and small scales typical in many placoderm lineages, the scales of *Entelognathus* are extraordinarily thin and large. Surprisingly, many assume a rhomboid shape and a subset of them possesses the peg-and-socket articulation previously considered a synapomorphy of bony fishes[23–28] (Fig. 5). Like the rhomboid scales of osteichthyans, some scales of *Entelognathus* also bear an anterodorsal process that is distinct from the dorsal peg, as well as a basal keel. The basal keel is less pronounced than most osteichthyan examples, which we attribute to the thinness of the scales in *Entelognathus*. Scales of *Entelognathus* are ornamented with longitudinally arranged ridges on the exposed field. Again, this is consistent with osteichthyan scales[23–28], and is not seen in placoderm scales, which mostly bear tubercular ornament[38,39]. However, the lateral line scales of *Entelognathus* are primitive with respect to osteichthyan scales, carrying the sensory line in an open groove rather than a buried canal[25,28,38,41,50–53] (Fig. 5).

Contrary to the characteristic three-layered structure in placoderm exoskeleton[38,54–56], the dermal plates and scales of *Entelognathus* lack a cancellar middle layer (Supplementary Fig. 5a–d), similar to early

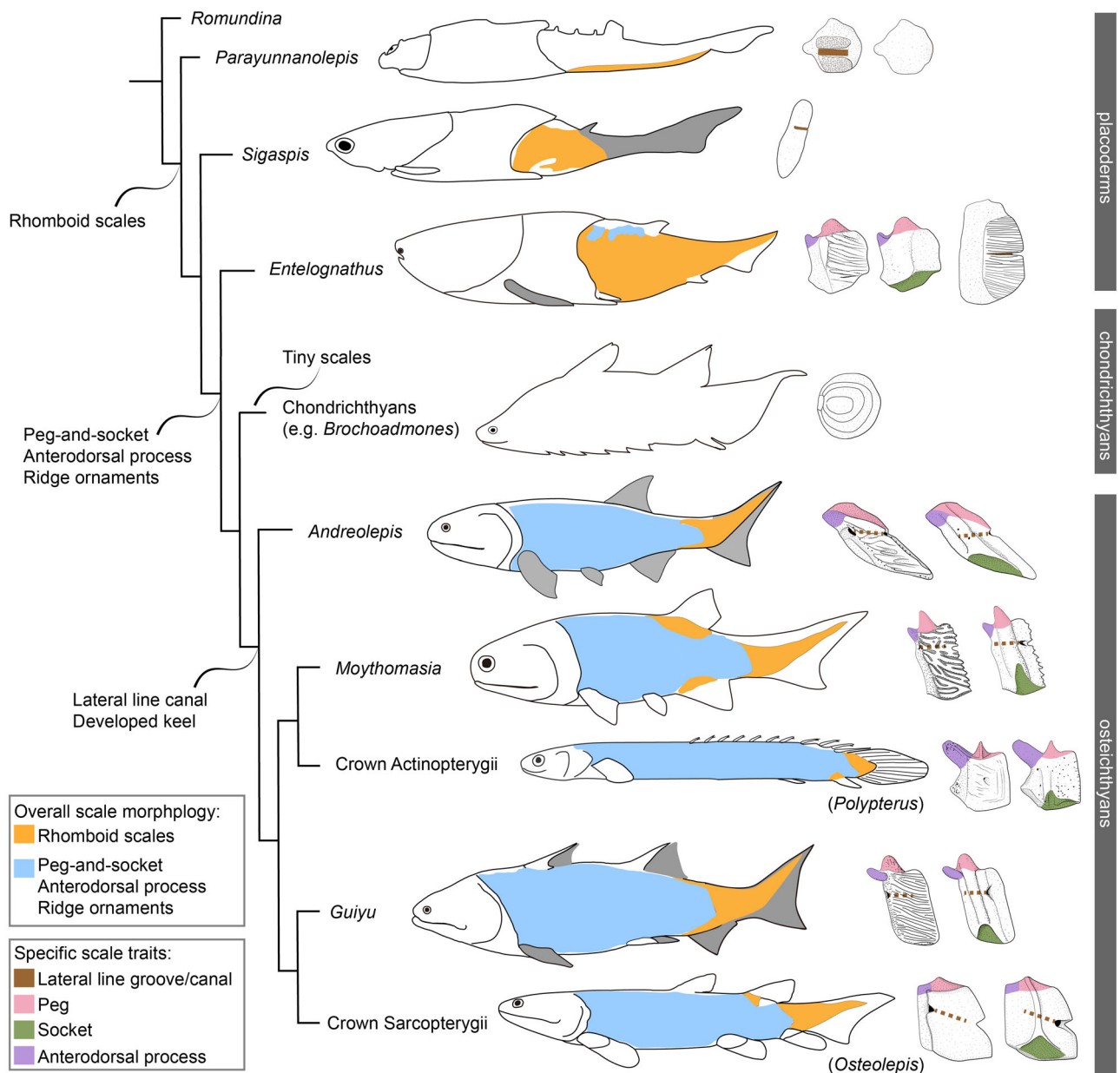

**Fig. 5 | Summary phylogeny of early jawed vertebrates showing the transformation in scale morphology and squamation.** This cladogram is simplified from the strict consensus tree of the 100,000 most parsimonious trees (Supplementary Fig. 6). The character transformations mapped on the tree are based on the results of phylogenetic analyses. Drawing sources: *Parayunnanplepis*[41]; *Sigaspis*[50]; *Brochoadmones*[13]; *Andreolepis*[52]; *Moythomasia*[46,94]; *Polypterus*[95]; *Guiyu*[28]; *Osteolepis*[45].

chondrichthyans[20,57–62]. The cancellar bone stratum containing developed vascular canals of early osteichthyans is also identified as a middle layer[41,55], yet these vascular canals extend into the superficial layer[25,33,34,63–66], unlike the condition in placoderms. Given the phylogenetic proximity of *Entelognathus* to the common ancestor of osteichthyans and chondrichthyans, the absence of the cancellar middle layer of the exoskeleton might be an ancestral crown-group character retained in chondrichthyans. The middle layer in the dermal plates and scales of osteichthyans is then acquired independently from that in the exoskeletons of the conventionally defined placoderms[23,41,55]. Alternatively, the loss of middle layer is independent in maxillate placoderms and chondrichthyans, with osteichthyans retaining the middle layer typical of most placoderms. In addition, the *Entelognathus* scales have distinctive vertical vascular canals extending through the superficial and basal layers (Supplementary Fig. 5a–d), reminiscent of the vertical vascular canals of the stem chondrichthyan

*Gualepis*[11]. Although heavy recrystallization obscures the tissue type of the superficial layer in *Entelognathus*, the thick basal layer composed of lamellar bone (Supplementary Fig. 5a–d) resembles that of early osteichthyans[25,27,33,34,63–66].

The osteichthyan-like peg-and-socket articulation of *Entelognathus* scales merits further consideration. Rhomboid scales are known for many primitive placoderms, such as the antiarch *Parayunnanolepis xitunensis*[41,67] and the arthrodires *Sigaspis lepidophora*[50] plus *Kujdanowiaspis podolica*[68], but all lack the peg-and-socket articulation. Perhaps significantly, scales with a peg-and-socket articulation show a restricted distribution in *Entelognathus*, and are limited to near the dorsal midline anterior to the dorsal fin. By contrast, peg-and-socket articulations characterize most flank scales of early osteichthyans[24–29], even in taxa that bear tiny scales[69] (Fig. 5). We hypothesize that the peg-and-socket articulation on rhomboid scales first appeared near the dorsal midline anterior to

the dorsal fin and gradually spread to other parts of the body prior to the origin of bony fishes. Like *Entelognathus*, scales on the tail of early osteichthyans lack pegs and sockets, and this perhaps can be interpreted as retention of a primitive condition. The lack of peg-and-socket articulation in scales on the tail might also indicate a functional constraint. The highly kinetic tail requires a substantial flexibility between the scales, perhaps accounting for the absence of the peg-and-socket articulation.

Given the phylogenetic proximity of *Entelognathus* to the common ancestor of osteichthyans and chondrichthyans, the presence of a peg-and-socket articulation between rhomboid scales might be an ancestral gnathostome character retained in osteichthyans. This trait would have been lost independently in chondrichthyans. Alternatively, the acquisition of peg-and-socket articulation of rhomboid scales occurred independently in *Entelognathus* and osteichthyans, with the absence of this articulation in chondrichthyans representing the retained primitive condition typical of most placoderms. While these two scenarios are equally parsimonious, the first hypothesis—the presence of rhomboid scales with peg-and-socket articulations in the last common ancestor of crown gnathostomes—is broadly consistent with the general pattern of reduction or simplification of many aspects of the chondrichthyan dermal skeleton. Better documentation of scale morphology for taxa adjacent to the gnathostome crown node will be essential to discriminating between these two scenarios. In addition, some phylogenetic analyses placed *Entelognathus* in the deepest position of the placoderm clade[70,71]. Based on these phylogenies, the peg-and-socket articulation of rhomboid scales might even be present in the common ancestor of gnathostomes, with the condition lost in core placoderms and chondrichthyans.

In addition, *Entelognathus* shows a complement of fin spines that adds significant details to the evolution of these structures. *Entelognathus*[9] bears a pair of dermal pectoral girdles with spinal plates, as in other placoderms[72–75], stem chondrichthyans[76–79], and some early osteichthyans[53,80,81] (Supplementary Fig. 7). *Entelognathus* also has a dorsal fin spine, which is absent in most placoderms[39] but present in some ptyctodonts[82–84]. The dorsal fin spine is common in chondrichthyans[76–79] and some early osteichthyans[53,80,81] (Supplementary Fig. 7). The dorsal fin spine (Fig. 2k) of *Entelognathus* is more similar to that of early osteichthyans, such as *Psarolepis*[80] and *Guiyu*[81], in terms of its robust basal plate and longitudinal ridges. Remarkably, the articulated specimen of *Entelognathus* exhibits an anal fin spine (Fig. 4a–c), which is unknown in any other placoderms. The anal fin spine and paired pelvic fin spines were previously thought to be restricted to stem chondrichthyans[76–79,85,86] and were regarded as two probable synapomorphies of the chondrichthyan total-group (Supplementary Fig. 7). The presence of an anal fin spine in *Entelognathus* suggests, however, that this character might have a deeper origin on the gnathostome stem.

Aside from the remarkable articulated specimens from China, the fossil record of Silurian jawed vertebrates is sparse and dominated by disarticulated scales, the identification of which is based on characters similar to the latter, articulated specimens of the morphologically well-separated major groups. The discovery of the peg-and-socket articulation in the stem gnathostome *Entelognathus* demonstrates that this character is no longer exclusive to—and diagnostic of—osteichthyans. Accordingly, extreme care must be taken using it to suggest osteichthyan affinity for isolated scales[86], although other features like an enclosed lateral line canal can be used to identify isolated scales as belonging to bony fishes. Our data on the postcranial exoskeleton of *Entelognathus* provide further evidence for the stepwise assembly of major features of osteichthyan anatomy before the last common ancestor of living jawed vertebrates, and further emphasize the profound specializations of the chondrichthyan dermal skeleton[9,87].

## Methods

### Fossil specimens

This study is based on one articulated specimen (IVPP V32322), two isolated plates (V32323.1 and V32323.2), and three isolated scales (V32323.3, V32323.4, V32323.5) of *Entelognathus primordialis*, and a lower jaw (V32324.1) plus one scale (V32324.2) of *Guiyu oneiros*, housed at the Institute of Vertebrate Paleontology and Paleoanthropology (IVPP), Chinese Academy of Sciences. They are collected from the muddy limestone from the Kuanti Formation (late Ludlow, Silurian) of Qujing, Yunnan Province, China, with the permission of Qujing government, following the national laws.

### X-ray computed microtomography and 3D reconstruction

The articulated specimens (IVPP V32322) and one isolated scale (V32323.5) were scanned with a GE phoenix v|tome|x m300&180 micro-computed tomography scanner at the IVPP. V32322 was scanned with an energy of 140 kV and a flux of 150 μA at a detector resolution of 20.500 μm per pixel. V32323.5 was scanned with an energy of 110 kV and a flux of 120 μA at a detector resolution of 2.589 μm per pixel. Tomographic data were segmented using Mimics (v.25.0, http://biomedical.materialise.com/mimics; Materialise), with images of models rendered in Blender[88].

### Geometric morphometric analysis

We selected the 3D virtual models of left flank scales to do the geometric morphometric analysis. We printed their images in crown view within Mimics 25.0. If a scale is missing or incomplete at one position, a mirror image of the right flank scale (if present) at the same position is used instead. 237 scales were selected (Supplementary Data 2). Then each scale outline was scaled before being digitized in a counterclockwise direction from a common starting location, the posteroventral corner, and saved as 50 equidistant semilandmark coordinate points (Supplementary Data 3) with TPSDig 2.32 software[89].

The coordinates were superimposed by Generalized Procrustes Analysis (GPA) using the R "geomorph" package for geometric morphometrics[90] (Supplementary Code 1). We conducted a principal component analysis (PCA) and the mean shape of each morphotype of the scales with R (Supplementary Code 1). The height, length, and aspect ratio of the scales were arranged in a matrix according to the natural position of the scales on the body and plotted with PAST 4.11[91] (Supplementary Data 4).

### Phylogenetic analysis

The character data entry and formation were performed in Mesquite 3.61[92]. A maximum-parsimony analysis was conducted in TNT 1.5[93]. All of the characters were unordered and unweighted. Osteostraci was set to be the outgroup. The analysis was conducted using a traditional search, with 100,000 maximum trees in memory and 1000 replicates of Wangner trees using random additional sequences. Bremer support values were generated in TNT, and Bremer decay indices retained suboptimal trees up to 20 extra steps. Our analysis generated 100,000 trees of 1861 steps (Consistency index = 0.374; retention index = 0.797), which are summarized as a strict consensus tree (Supplementary Fig. 6a) and a 50% majority-rule consensus tree (Supplementary Fig. 6b).

### Reporting summary

Further information on research design is available in the Nature Portfolio Reporting Summary linked to this article.

## Data availability

The CT data and 3D models generated in this study have been deposited in the figshare database under the accession code: https://figshare.com/s/f388c2c162e962aab711. All other data generated in this study are provided in the Supplementary Information Data file. The fossil specimens are housed at the Institute of Vertebrate Paleontology

and Paleoanthropology, Chinese Academy of Sciences. They are available on request.

## Code availability

The codes used to perform geometric morphometric analysis are available in Supplementary Code 1.

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

## Acknowledgements

We thank L.T. Jia for photograph; C.Y. Xiong for specimen preparation; Y.M. Hou and P.F. Yin for CT scanning; L.J. Peng for assistance with making thin sections; H.M. Zhang, S.M. Dai, and B. Yang of NICE Vistudio (Paleovislab, IVPP) for designing and restoring 3D models; Y.L. Sun for discussion. This work was supported by the Open Research Program of the International Research Center of Big Data for Sustainable Development Goals (CBAS2022ORP01, M.Z., Y.Z., and X.C.), the National Natural Science Foundation of China (42130209, 42302005, and 42272028, M.Z., Y.Z., X.C., and Y.Y.), the Strategic Priority Research Program of Chinese Academy of Sciences (XDA19050102 and XDB26000000, M.Z., Y.Z., and X.C.), the Chinese Postdoctoral Science Foundation grant (2022M720215, X.C.), and the State Key Laboratory of Palaeobiology and Stratigraphy grant (No. 223106, X.C.).

## Author contributions

M.Z. and Y.Z. designed the project. X.C. collected tomographic data and processed the tomographic data. X.C. and Y.Z. assembled and analyzed the phylogenetic dataset. X.C. and Y.Y. performed the Geometric Morphometric Analyses. X.C., M.F., Y.Z., and M.Z. produced the figures. X.C. interpreted the data and wrote the first draft of the manuscript. X.C., M.F., Y.Z., and M.Z. discussed and commented on the final version of the manuscript.

## Competing interests

The authors declare no competing interests.
