## [Peer Review File · Nature Communications]

Bony-fish-like scales in a Silurian maxillate placodermReviewers' Comments:

Reviewer #1:

Remarks to the Author:

Review of Cui et al. manuscript "Bony-fish-like scales in a Silurian maxillate placoderm"

Noteworthy results of this project include the identification of a transitional stage between scales of stem gnathostomes ('placoderms') and osteichthyans in the maxillate 'placoderm' *Entelognathus*, supporting the previous inference of this transitional position based on its jaws and dentition. The use of CT scanning to study the morphology of every individual dermal skeletal element (scales, plates, and fin spines) of this taxon is particularly illuminating. Another interesting result is the identification of median fin spines, in particular anal spines, the latter otherwise only known from stem chondrichthyans.

The work is very significant for all work on early vertebrates, highlighting a surprising combination of characters previously considered diagnostic for class-level taxonomic groups. The analysis and interpretation of structures supports the conclusions of the authors; the methodology is sound and the work is of a high standard, and could be reproduced based on the data provided. Character codings and nomenclature for some taxa (highlighted) in the supplementary material could be updated based on more recent publications than those listed. Additional references include: Burrow et al. 2015 (*Climatius reticulatus*), Burrow & Young 2012 (*Culmacanthus stewarti*), Burrow et al. 2016 (*Diplacanthus crassissimus* = *Diplacanthus striata*), Newman et al. 2012 (*Uraniacanthus probation* = *Gladobranchus probaton*), Burrow et al. 2018 (*Ischnacanthus gracilis*), Burrow et al. 2023 (*Ligulalepis toombsi*), Burrow et al. 2022 (*Mesacanthus mitchelli*), Vergoossen 1997 (*Zemlyacanthus menneri* = *Poracanthodes menneri*), Dearden et al. 2021 (*Dobunnacanthus waynensis* = *Vernicomacanthus waynensis*).

I would like to see more added to the discussion, which now focusses on the scales. Please add something about the structure and significance of the fin spines identified on the animal.

I have made some comments on the annotated manuscript.

Carole Burrow, 2nd July 2023

Reviewer #2:

Remarks to the Author:

Dear authors,

Your work is of fundamental importance as it is changing our view on the evolution of jawed vertebrates. With the description of scales that have peg-and-socket articulations, characters that are thought to be synapomorphies for osteichthyans you provide important evidence for the phylogenetic position of *Entelognathus primordialis* and for the evolution of vertebrate characters.

Your work has significant impact on palaeobiology, evolutionary biology and larger implications for biodiversity research discussing the acquisition of novelties, diversity and disparity. This analysis of scales is novel to the scientific literature and the methods are of high quality.

The conclusions are all well supported by the data and analyses, which is performed following highest standards.

Your methodology is sound and there is enough detail provided to reproduce your work.

Nevertheless I have some suggestions to improve your manuscript.

Line 40, page 2: Use always the same terminology in the text when using "placoderm" continue doing so in the text as you indicate a paraphyletic group, or placoderm in case you consider a monophyletic group. This would simplify reading your text.

Line 200, page 10: In lateral view the canals appear in different levels, with a concentration dorsally and more elongated ones ventrally. Are they all connected as indicated here or of different generations indicating appositional growth of the tubercles? This is not clearly visible in the figures. Please indicate the bifurcation you mention in the text also in the figure, this is not obvious for me.

Line 300, page 14: The lack of pegs and sockets in scales on the tail might also indicate a functional constraint. Please discuss that possibility.

Line 613, following, page 28/29, Figure 1: Please add the explanations of the annotations in the figure captions (Fig. 1a, b).

Recommendations to improve the supplementary information:

Supplementary figure 1, 3 and 4: Please add the explanations of the annotations in the figure captions.

Reviewer #3:

Remarks to the Author:

I have certainly enjoyed reviewing this paper, which is well written and structured. The manuscript provides the first description of Entelognathus post-thoracic exoskeleton, which has emerged as a key taxon for understanding phylogenetic issues of early gnathostome evolution. The anterior part this "maxillate" placoderm was known to combine a mosaic of characters associated with "placoderms" or crown gnathostomes. Here, the authors show that the postcranial exoskeleton in Entelognathus also combines mosaic elements, i.e. scales similar to that of osteichthyans along with some characteristic fin spines of stem chondrichthyans. I believe that the mns. could be of great interest to a wide audience. As such, I recommend that it be accepted with some consideration.

I reach agreement with the part concerning with the scales. The descriptions and figures are excellent. I fully agree with the evolutionary interpretations and the proposed transformation in scale morphology and squamation (Discussion, Figure 4). This part corresponds to the bulk of the work and fits perfectly with the title and the main conclusions of the article.

I am not clear in the objectives of the performed geometric morphometrics analysis...It is just to quantitative differentiate the scales morphotypes??? In that case, most of the identified morphotypes seems to overlap in the analysis (Fig 1c Supplementary fig 2). Anyway only a brief mention on this laborious analysis is made in the main text.

My major concerns fall on the part of the spines. The authors named, in the summary, and describe, in the results, the presence of several unpaired spines and a pectoral spine. However, no mention is made of them in the discussion or in evolutionary interpretations (e.g. Figure 4).

In fact, according to the figures provided, these spines could be confused with modified median dorsal scales or "fulcral" scales present in numerous groups of primitive vertebrates, for example placoderms (see for example Burrow, C. J. & Turner, S. 1999. figure 2, H) and stem-osteichthyans (eg *Andreolepis* see Chen, 2010) and they look distinctly different from "acanthodian" spines.

Therefore, I would suggest the authors clarify exactly how they identify these structures as spines and differentiate them from modified median scales. Especially since he considers them homologous to those present in the "acanthodians".

In any case, multiple (unpaired) anal spines are certainly very rare, even in "acanthodians" (in some *Acanthodes*?). So the authors should be clear on this point (which group of acanthodians have multiple anal fins?) and put it into a phylogenetic contest.

Authors need check figures and figure captions. No available anatomical abbreviations. No scales in figure 2 and 3. Pectoral spine is not included in Figure 2

HB

We thank the referees for their constructive comments on the manuscript, which we feel have helped us produce a much-improved paper.

Reviewer #1 (Remarks to the Author):

Character codings and nomenclature for some taxa (highlighted) in the supplementary material could be updated based on more recent publications than those listed. Additional references include: Burrow et al. 2015 (*Climatius reticulatus*), Burrow & Young 2012 (*Culmacanthus stewarti*), Burrow et al. 2016 (*Diplacanthus crassisimus* = *Diplacanthus striata*), Newman et al. 2012 (*Uraniacanthus probation* = *Gladobranthus probation*), Burrow et al. 2018 (*Ischnacanthus gracilis*), Burrow et al. 2023 (*Ligulalepis toombsi*), Burrow et al. 2022 (*Mesacanthus mitchelli*), Vergoossen 1997 (*Zemlyacanthus menneri* = *Poracanthodes menneri*), Dearden et al. 2021 (*Dobunnacanthus waynensis* = *Vernicomacanthus waynensis*).

[Response]: We updated some character codings and nomenclature of the taxa mentioned above in the matrix (Supplementary Data 1), and re-analyzed phylogenetic relationships on the basis of the updated data. The revised codings and nomenclature are as follows:

Climatius reticulatus (Burrow et al. 2015): The reference added.

Culmacanthus stewarti (Burrow and Young 2012): The reference added.

Diplacanthus crassisimus: Adopted. We use the valid name *Diplacanthus crassisimus* to replace *Diplacanthus striata* that is a synonym of the former (Burrow et al. 2016).

Doliodus latispinosus = *Doliodus problematicus*: Adopted. We use the valid name *Doliodus latispinosus* to replace *Doliodus problematicus* that is a synonym of the former (Burrow et al. 2023a). Characters 13 and 16 are coded from ? to 0.

Uraniacanthus probation = *Gladobranthus probation*: Adopted. We use the valid name *Uraniacanthus probation* to replace *Gladobranthus probation* that is a synonym of the former (Newman et al. 2012).

Ischnacanthus gracilis (Burrow et al. 2018): The reference added.

Ligulalepis toombsi (Burrow et al. 2023b)

Character 10: Dentine tissue. ? to 1 (present)

Character 11: Dentine kind. ? to 2 (orthodentine)

Character 13: Enamel(oid) present on dermal bones and scales. ? to 1 (present)

Character 14: Enamel. ? to 0 (single-layered)

Character 15: Enamel layers. ? to 1 (separated by layers of dentine)

Character 16: Enamel(oid) on teeth. ? to 1 (present)

Character 25: Bone cell lacunae in body scale bases. ? to 0 (present)

Character 27: Lepidotrichia or lepidotrichia-like scale alignment ? to 0 (present)

Character 28: Differentiated lepidotrichia ? to 1 (present)

Character 32: Scute-like ridge scales (basal fulcra) ? to 1 (present)

Character 237: Lateral gular plates. ? to 1 (present)

Character 239: Median gular. ? to 0 (present)

Character 240: Oral dermal tubercles borne on jaw cartilages. ? to 1 (present)

Character 254: Number of fang pairs on posterior coronoid. ? to 0 (one)

Character 267: Premaxilla. ? to 0 (extends under orbit)

Character 469: Lateral cranial canal. ? to 1 (present)

Character 558: Macromeric dermal shoulder girdle. ? to 0 (present)

Dialipina salgueiroensis (Burrow et al. 2023b)

Character 11: Dentine kind. ? to 2 (orthodentine)

Mesacanthus mitchelli (Burrow et al. 2022)

Character 36: Body scale growth concentric. ? to 1 (present)

Character 37: Body scales with peg-and-socket articulation. ? to 0 (absent)

Character 40: Body scales with bulging base. ? to 1 (present)

Character 45: Neck canal. ? to 0 (absent)

Character 46: Keel of scale. ? to 0 (absent)

Character 47: Posterior ledge (secondary keel) of scale. ? to 0 (absent)

Character 48: Anteroventral process on scale. ? to 0 (absent)

Character 49: Ventral process of scale. ? to 1 (absent)

Character 50: Anterodorsal process on scale. ? to 0 (absent)

Character 54: Longitudinal scale alignment in fin webs. ? to 1 (absent)

Character 674: Pelvic fin spine. ? to 1 (present)

Character 691: Shape of trunk scales. ? to 1 (small/round)

Character 692: Dermal ornament with parallel vermiform ridges on trunk scales. ? to 0 (absent)

Character 693: Dermal ornament with concentric ridges on trunk scales. ? to 0 (absent)

Character 694: Dermal ornament with tubercles on trunk scales. ? to 0 (absent)

Zemlyacanthus menneri = *Poracanthodes menneri*: Adopted. We use the valid name *Zemlyacanthus menneri* to replace *Poracanthodes menneri* that is a synonym of the former (Vergoossen 1997).

Dobunnacanthus waynensis = *Vernicomacanthus waynensis*: Adopted. We use the valid name *Dobunnacanthus waynensis* to replace *Vernicomacanthus waynensis* that is a synonym of the former. Some codings were updated based on the new reference (Dearden et al. 2021).

Character 35: Body scale growth pattern. ? to 1 (polyodontode)

Character 691: Shape of trunk scales. ? to 1 (small/round)

Character 692: Dermal ornament with parallel vermiform ridges on trunk scales. ? to 0 (absent)

Character 693: Dermal ornament with concentric ridges on trunk scales. ? to 0 (absent)

Character 694: Dermal ornament with tubercles on trunk scales. ? to 0 (absent)

Rhadinacanthus longispinus (Burrow et al., 2016)

Character 11: Dentine kind. ? to 0 (mesodentine)

Character 20: Resorption and redeposition of odontodes. ? to 0 (lacking or partially developed)

Character 27: Lepidotrichia or lepidotrichia-like scale alignment. ? to 0 (present)

Character 33: Flank scales alignment. ? to 1 (oblique rows or hexagonal/rhombic packing)

Character 34: Scales. ? to 1 (micromeric)

Character 36: Body scale growth concentric. ? to 1 (present)

Character 37: Body scales with peg-and-socket articulation. ? to 0 (absent)

Character 39: Body scale profile. ? to 0 (distinct crown and base demarcated by a constriction (neck))

Character 40: Body scales with bulging base. ? to 1 (present)

Character 41: Body scales with flattened base. ? to 0 (absent)

Character 43: Profile of scales with constriction between crown and base. ? to 1 (neck greatly constricted, resulting in anvil-like shape)

Character 59: Dermal skull roof. ? to 1 (consists of undifferentiated plates or tesserae)

Character 60: Tesserae morphology. ? to 0 (large interlocking polygonal plates)

Character 61: Extent of dermatocranial cover. ? to 0 (complete)

Character 185: Jugal portion of infraorbital canal joins supramaxillary canal. ? to 0 (present)

Character 211: Consolidated cheek plates. ? to 1 (present)

Character 237: Lateral gular plates. ? to 0 (absent)

Character 243: Dermal jaw plates on biting surface of jaw cartilages. ? to 1 (present)

Character 245: Dermal plates on mesial (lingual) surfaces of Meckels cartilage and palatoquadrate. ? to 0 (absent)

Character 279: Median dermal bone of palate (parasphenoid). ? to 0 (absent)

Character 314: Enlarged adsymphysial tooth whorl. ? to 0 (absent)

Character 342: Pronounced dorsal process on Meckelian bone or cartilage. ? to 0 (absent)

Character 348: Interhyal. ? to 0 (absent)

Character 376: Nasal opening(s). ? to 1 (ventral and anterior to orbits)

Character 387: Prominent pre-orbital rostral expansion of the neurocranium. ? to 1 (absent)

Character 558: Macromeric dermal shoulder girdle. ? to 0 (present)

Character 565: Dermal shoulder girdle composition. ? to 1 (ventral components only)

Character 566: Dermal shoulder girdle forming a complete ring around the trunk. ? to 1 (absent)

Character 567: Pectoral fenestra completely encircled by dermal shoulder armour. ? to 1 (absent)

Character 619: Scapular process with posterodorsal angle. ? to 0 (absent)

Character 621: Ventral margin of separate scapular ossification. ? to 1 (deeply angled)

Character 623: Endoskeletal postbranchial lamina on scapular process. ? to 0

(present)

Character 624: Mineralisation of internal surface of scapular blade. ? to 0

(mineralised all around)

Character 625: Coracoid process. ? to 0 (absent)

Character 627: Paired (pectoral) fins. ? to 1 (present)

Character 642: Pelvic girdle with substantial dermal component. ? to 1 (no)

Character 691: Shape of trunk scales. ? to 1 (small/round)

Character 692: Dermal ornament with parallel vermiform ridges on trunk scales. ? to 1 (present)

Character 693: Dermal ornament with concentric ridges on trunk scales. ? to 0 (absent)

Character 694: Dermal ornament with tubercles on trunk scales. ? to 0 (absent)

I would like to see more added to the discussion, which now focusses on the scales. Please add something about the structure and significance of the fin spines identified on the animal.

[Response]: We added a paragraph in discussion section to discuss the evolution of fin spines in early jawed vertebrates. In addition, we added a figure (Supplementary Fig. 6) to show the transformation of fin spines in early jawed vertebrates.

Reviewer #2 (Remarks to the Author):

Line 40, page 2: Use always the same terminology in the text when using "placoderm" continue doing so in the text as you indicate a paraphyletic group, or placoderm in case you consider a monophyletic group. This would simplify reading your text.

[Response]: Adopted. In the text, we consistently use placoderms for simplicity whether the term represents a monophyletic or paraphyletic group, following the conventional usage in recent literature.

Line 200, page 10: In lateral view the canals appear in different levels, with a concentration dorsally and more elongated ones ventrally. Are they all connected as indicated here or of different generations indicating appositional growth of the tubercles? This is not clearly visible in the figures. Please indicate the bifurcation you mention in the text also in the figure, this is not obvious for me.

[Response]: Revised and augmented figures of the scale in different views are now added. We rechecked the CT images of the scale (V32323.5) and found that some vascular canals located in the ventral part were not reconstructed because they were filled with recrystallized minerals (Fig. 3f–h). Then we reconstructed the whole picture of the canal system of the scale to the extent possible (Figure 3a–e). In crown view, the canal system consists of scattered ascending canals that are mainly concentrated in the middle and posterior parts of the scale (Fig. 3a–e). All ascending canals are not connected. The whole canal system is semicircular, and the ascending canals incline to the center and are arranged in radial lines. In the lateral view, the ascending canals

appear at the same level, but they become shorter from anterior to posterior. This is because the scale becomes thinner posteriorly. In addition, it may be related to the supposed appositional growth of the scale, with the younger ascending canals lying posteriorly. The central ascending canals that represent the oldest canals are forked (Fig. 3b–e).

Line 300, page 14: The lack of pegs and sockets in scales on the tail might also indicate a functional constraint. Please discuss that possibility.

[Response]: We added the sentence “The lack of peg-and-socket articulations in scales on the tail might also indicate a functional constraint. The highly kinetic tail requires a substantial flexibility between the scales, perhaps accounting for the absence of peg-and-socket articulations.”

Line 613, following, page 28/29, Figure 1: Please add the explanations of the annotations in the figure captions (Fig. 1a, b).

[Response]: Added.

Supplementary figure 1, 3 and 4: Please add the explanations of the annotations in the figure captions.

[Response]: Added.

Reviewer #3 (Remarks to the Author):

I am not clear in the objectives of the performed geometric morphometrics analysis...It is just to quantitative differentiate the scales morphotypes??? In that case, most of the identified morphotypes seems to overlap in the analysis (Fig 1c Supplementary fig 2). Anyway only a brief mention on this laborious analysis is made in the main text.

[Response]: We performed the geometric morphometric analysis for two purposes. First, it helps to classify the scales together with the qualitative morphological differences. Second, it more clearly summarizes the degree of morphological variation between different morphotypes of scales through PCA scatter plots. Some morphotypes overlap in Fig. 1c because they are similar under the pc1 and pc2, but they overlap less in the 3D PCA plot of the first three principal components (Supplementary fig. 2a). Another reason for the partial overlap is that some scales in the overlapping areas of the plots have close natural positions on the fish body, so they appear similarities in morphology.

My major concerns fall on the part of the spines. The authors named, in the summary, and describe, in the results, the presence of several unpaired spines and a pectoral spine. However, no mention is made of them in the discussion or in evolutionary interpretations (e.g. Figure 4).

[Response]: We added a paragraph in discussion section to discuss the evolution of fin spines in early jawed vertebrates. In addition, we added a figure (Supplementary

Fig. 6) to show the transformation of fin spines in early jawed vertebrates.

In fact, according to the figures provided, these spines could be confused with modified median dorsal scales or “fulcral” scales present in numerous groups of primitive vertebrates, for example placoderms (see for example Burrow, C. J. & Turner, S. 1999. figure 2, H) and stem-osteichthyans (eg *Andreolepis* see Chen, 2010) and they look distinctly different from “acanthodian” spines. Therefore, I would suggest the authors clarify exactly how they identify these structures as spines and differentiate them from modified median scales. Especially since he considers them homologous to those present in the “acanthodians”.

In any case, multiple (unpaired) anal spines are certainly very rare, even in “acanthodians” (in some *Acanthodes*?). So the authors should be clear on this point (which group of acanthodians have multiple anal fins?) and put it into a phylogenetic contest.

[Response]: Revised. We identified the previous “posterior dorsal fin spine” and “anterior anal fin spine” as the fourth post median dorsal scale and median ventral scale, through careful re-examination of their morphology and comparisons with other placoderms, acanthodians, and some early osteichthyans. The evidence is as follows: First, there are no small fin scales behind them, refuting the existence of the posterior dorsal fin and anterior anal fin, and thus refuting their spine properties. Second, their overlapped areas on the external and internal surfaces indicate that they overlap with the surrounding flank scales, and thus have no or extremely weak free-ends. Third, no gnathostome has been reported to bear two anal fin spines. *Acanthodes* has an unpaired ventral spine between the pectoral and anal fin spines (Heidtke, 1990), but it is regarded as possible homologue of the pelvic fin spines (Beznosov, 2009).

The previous “anterior dorsal fin spine” and “posterior anal fin spine” are confidently identified as median fin spines, because both are immediately followed by small scales representing the bodies of the dorsal and anal fins, respectively. Both have morphologies consistent with spines, in that they have a basal plate covered by adjacent flank scales and a free protruding spinal portion.

Authors need check figures and figure captions. No available anatomical abbreviations. No scales in figures 2 and 3.

[Response]: Added.

Pectoral spine is not included in Figure 2.

[Response]: We checked the manuscript and confirmed that we did not describe the pectoral spine in the “results” section. The present specimen (IVPP V32322) did not preserve the dermal shoulder girdles.

Reviewers' Comments:

Reviewer #1:

Remarks to the Author:

One of the main results of this project is the recognition of a transitional stage between scales of stem gnathostomes ('placoderms') and osteichthyans in the maxillate 'placoderm' Entelognathus, supporting the previous inference of this transitional position based on its jaws and dentition. The other important result is the identification of median fin spines, in particular anal spines, the latter otherwise only known from stem chondrichthyans.

This original work is of interest to all early vertebrate workers, with the authors' conclusions supported by their methodology and results.

The authors have addressed all the issues raised by reviewers of the initial submission, revising their interpretations of some elements and expanding the Discussion to include relevant data on the evolution and nature of fin spines as well as scales. Description of the open lateral line canal should be added somewhere in the main text scale description, rather than just in the discussion and suppl.

A few minor grammatical corrections could still be made (line numbers on clean version of revision):

l. 17: 'year' for 'years'

l. 19: 'was previously' for 'is'

l. 26: 'illustrates that many' for 'illustrates many'

l. 50: 'Many early chond...' for 'Chond...': [the 'acanthodian' taxa more closely related to crown Chondrichthyes (Climatius, Dobunnacanthus, Parexus etc) don't have bulging bases, it is the more conventional acanthodian taxa (Acanthodes, Ischnacanthus, etc) that do]

l. 53: 'Some early chondrichthyan' for 'chondrichthyan': [again, Climatius etc do not have enameloid]

l. 74: 'fin spine' for 'fin-spine'

l. 80: 'part' for 'the part '

l. 139: ?replace 'lingual' with a different descriptor

l. 355: 'the new' for 'new'

l. 359: 'and were regarded as two synapomorphies' for 'and two probable synapomorphies'

It has been a pleasure to review this work.

Carole Burrow. 14th September 2023

Reviewer #2:

Remarks to the Author:

Dear authors,

I am delighted to see your revised manuscript improved including the reviewers comments.

Your manuscript is an important contribution demonstrating the presence of characters in the stem of gnathostomes also in the postcranial skeleton that are traditionally considered as synapomorphies of osteichthyans. Therefore, these are additional examples of an early evolution of traits and secondary simplifications of these traits in chondrichthyans.

Your work is significant for evolutionary biology and is changing textbook views on vertebrate evolution.

Your argumentation improved, especially including the up to date taxonomy and character coding. The additional discussion on the presence and evolution of spines make it a well supported manuscript.

Thanks for the detailed information on the histology and argument for the development of the scale and consideration of functional constraints.

I recommend publication of your manuscript.

However I have some minor comments.

Typo in Line 319 -320: reminiscent of the vertical vascular canals of the stem chondrichthyan Gualepis.

- correct chondrichthyan into chondrichthyan.

Please change in line 381: the identification of which is based on characters similar to the later, Should be latter.

Additional comment to include:

In the discussion your argumentation on the evolution of the peg-and-socket articulation of rhomboid scales (line 347-353) is based on your parsimony analysis:

You argue for two hypotheses: "the presence of a peg-and-socket articulation between rhomboid scales might be an ancestral gnathostome character retained in osteichthyans. This trait would have been lost independently in chondrichthyans. Alternatively, the acquisition of peg-and-socket articulation of rhomboid scales occurred independently in Entelognathus and osteichthyans, with the absence of this articulation in chondrichthyans representing the retained primitive condition typical of most placoderms."

As there is a debate on the ancestral state and phylogenies with placoderms being monophyletic or paraphyletic the interpretation of your data could also change. Using a Bayesian approach the phylogenies were interpreted very differently including the position of Entelognathus in the tree.

Please mention the uncertainties and in case add some citations:

Rücklin, M., King, B., Cunningham, J.A., Johanson, Z., Marone, F., Donoghue, P.C.J. 2021. Acanthodian dental development and the origin of gnathostome dentitions. *Nature Ecology and Evolution*.

<https://doi.org/10.1038/s41559-021-01458-4>

King, B., Rücklin, M. 2020. A Bayesian approach to dynamic homology of morphological characters and the ancestral phenotype of jawed vertebrates. *eLife*, 9: e62374. <https://doi.org/10.7554/eLife.62374>

Based on these phylogenies the peg-and-socket articulation of rhomboid scales might even be present in the common ancestor of (apomorphy defined) gnathostomes. With the condition lost in core placoderms and acanthodians/chondrichthyans.

Data sharing: Thanks for sharing the 3D virtual models of IVPP V32322 on figshare. Ideally, you would also upload the tomographic dataset of the articulated specimen IVPP V32322 and isolated scale V32323.5 on a repository to ensure open data principles and reproducibility of your research.

Reviewer #3:

Remarks to the Author:

I think that the authors have taken into account all the comments of the referees/editors in this new version of the manuscript.

They respond carefully and in great detail to all the questions raised in the review.

I recommend publication of the article in *Nature Communications*

We thank the referees for their constructive comments on the manuscript, which we feel have helped us produce a much-improved paper.

Reviewer #1 (Remarks to the Author):

Description of the open lateral line canal should be added somewhere in the main text scale description, rather than just in the discussion and suppl.

[Response]: Added (lines 137–142).

A few minor grammatical corrections could still be made (line numbers on clean version of revision):

l. 17: 'year' for 'years'

[Response]: Revised.

l. 19: 'was previously' for 'is'

[Response]: Revised.

l. 26: 'illustrates that many' for 'illustrates many'

[Response]: Revised.

l. 50: 'Many early chond...' for 'Chond...': [the 'acanthodian' taxa more closely related to crown Chondrichthyes (Climatius, Dobunnacanthus, Parexus etc) don't have bulging bases, it is the more conventional acanthodian taxa (Acanthodes, Ischnacanthus, etc) that do]

[Response]: Revised.

l. 53: 'Some early chondrichthyan' for 'chondrichthyan': [again, Climatius etc do not have enameloid]

[Response]: Revised.

l. 74: 'fin spine' for 'fin-spine'

[Response]: Revised.

l. 80: 'part' for 'the part'

[Response]: Revised.

l. 139: ?replace 'lingual' with a different descriptor

[Response]: Revised. We replace 'lingual' with 'lingulate'.

l. 355: 'the new' for 'new'

[Response]: Revised.

l. 359: 'and were regarded as two synapomorphies' for 'and two probable

synapomorphies'

[Response]: Revised.

Reviewer #2 (Remarks to the Author):

Typo in Line 319 -320: reminiscent of the vertical vascular canals of the stem chondrichthyan *Gualepis*. - correct chondrichthyan into chondrichthyan.

[Response]: Corrected.

Please change in line 381: the identification of which is based on characters similar to the later,

Should be latter.

[Response]: Corrected.

Additional comment to include:

In the discussion your argumentation on the evolution of the peg-and-socket articulation of rhomboid scales (line 347-353) is based on your parsimony analysis: You argue for two hypotheses: "the presence of a peg-and-socket articulation between rhomboid scales might be an ancestral gnathostome character retained in osteichthyans. This trait would have been lost independently in chondrichthyans. Alternatively, the acquisition of peg-and-socket articulation of rhomboid scales occurred independently in *Entelognathus* and osteichthyans, with the absence of this articulation in chondrichthyans representing the retained primitive condition typical of most placoderms."

As there is a debate on the ancestral state and phylogenies with placoderms being monophyletic or paraphyletic the interpretation of your data could also change. Using a Bayesian approach the phylogenies were interpreted very differently including the position of *Entelognathus* in the tree. Please mention the uncertainties and in case add some citations:

Rücklin, M., King, B., Cunningham, J.A., Johanson, Z., Marone, F., Donoghue, P.C.J. 2021. Acanthodian dental development and the origin of gnathostome dentitions.

Nature Ecology and Evolution. <https://doi.org/10.1038/s41559-021-01458-4>

King, B., Rücklin, M. 2020. A Bayesian approach to dynamic homology of morphological characters and the ancestral phenotype of jawed vertebrates. *eLife*, 9: e62374. <https://doi.org/10.7554/eLife.62374>

Based on these phylogenies the peg-and-socket articulation of rhomboid scales might even be present in the common ancestor of (apomorphy defined) gnathostomes. With the condition lost in core placoderms and acanthodians/chondrichthyans.

[Response]: Added (lines 354–359).

Data sharing: Thanks for sharing the 3D virtual models of IVPP V32322 on figshare. Ideally, you would also upload the tomographic dataset of the articulated specimen

IVPP V32322 and isolated scale V32323.5 on a repository to ensure open data principles and reproducibility of your research.

[Response]: Added. We uploaded the tomographic dataset of IVPP V32322 and V32323.5 to figshare database under the accession code:

<https://figshare.com/s/f388c2c162e962aab711>.